



# Unlocking the Full Potential of Wake Steering: Implementation and Assessment of a Controls-Oriented Model

Christopher J. Bay[1], Jennifer King[1], Paul Fleming[1], Rafael Mudafort[1], and Luis A. Martínez-Tossas[1]

[1]National Wind Technology Center, National Renewable Energy Laboratory, Golden, CO, 80401, USA

*Correspondence to:* Christopher Bay (christopher.bay@nrel.gov)

**Abstract.** In this work, a controls-oriented wake model is modified and compared to an analytical Gaussian wake model and high-fidelity simulation data. This model, called the curled wake model, captures a wake phenomenon that occurs behind yawed turbines, modeled as a collection of vortices shed from the rotor plane. Through turbine simulations, these vortices are shown to have a significant impact on the prediction of wake steering's performance. Also, optimizations using the model are performed and produce results consistent with recent published research. Results indicate that wind farm controllers designed and analyzed with the curled wake model produce wake steering controllers which can realize larger gains in power production than previously estimated. Overall, the results support the concept of secondary steering, or a yawed turbine's ability to deflect the wake of a downstream turbine, and suggest that future turbine wake studies and yaw optimizations should include the curled wake phenomenon.

## 1 Introduction

Flow control in wind power plants aims to manipulate a turbine's influence on the wind to achieve an increase in performance at the plant level. One method of flow control known as wake steering intentionally misaligns turbines from the incoming flow to deflect their wakes away from downstream turbines (Wagenaar et al., 2012; Adaramola and Krogstad, 2011; Park et al., 2013; Gebraad et al., 2016). Research has shown that with certain misalignments the overall production of a wind plant can increase, even though the misaligned turbines experience an individual power loss (Dahlberg and Medici, 2003).

Numerous techniques have been used to study wake steering, including theoretical, experimental, and field campaigns. Fleming et al. (2015) observed in computational fluid dynamics (CFD) simulations an overall increase in power production of an offshore wind turbine array of two turbines using wake steering. Large eddy simulations (LES) by Vollmer et al. (2016) were used to investigate the feasibility of wake steering under different atmospheric conditions, finding that stable atmospheric conditions provide for the largest performance increases with wake steering. Howland et al. (2016) characterized the shape of yawed turbine wakes in a porous disk experiment as well as LES.

Wake deflection due to yaw misalignment has also been studied in wind tunnel experiments (Medici and Alfredsson, 2006; Schottler et al., 2017; Bartl et al., 2018). Park et al. (2016) conducted an optimal yaw angle study for six scaled turbines in a wind tunnel using a data-driven Bayesian Ascent method. The results showed that the optimal yaw settings resulted in the progressively smaller yaw angles for turbines that were further downstream. Additionally, a handful of field campaigns





have been conducted to evaluate the effectiveness of wake steering (Wagenaar et al., 2012; Fleming et al., 2017). Fleming et al. (2019) found a 13% increase in energy on a downstream turbine for two closely spaced turbines within a 10° bin, consistent with prior predictions from different models. Howland et al. (2019) leveraged site-specific, historical operational data to develop a wake control scheme that was tested in a land-based wind power plant in Alberta, Canada. The controller resulted in power increases of 7%–47% for wind conditions that commonly occur at night, as well as a decrease in variability of power production of up to 72%.

Alongside theoretical and experimental studies, engineering models have been crafted to predict wake steering and assess its overall benefit. One controls-oriented tool used in the design and study of turbine wake controls is the FLOw Redirection and Induction in Steady State (FLORIS) model (Gebraad et al., 2016). Originally based on the Jensen (Jensen, 1984) and Jiménez (Jiménez et al., 2010) models, FLORIS has evolved into a software repository that contains several wake models. More recently, FLORIS uses the wake recovery and redirection models of Bastankhah and Porté-Agel (2014, 2016), and Niayifar and Porté-Agel (2015). Further description of FLORIS and its models is covered in the next section, but the reader is referred to Annoni et al. (2018) for a detailed description of the current FLORIS models. The FLORIS model is open source and the latest version is available for download and collaborative development (https://github.com/NREL/FLORIS).

These models have shown increasing ability to accurately design wake steering controllers for the case of one turbine waking a second; however, their success in modeling larger arrays of turbines implementing wake steering is less established. For one reason, there is no field data for this case, and although there are some LES studies, it is expensive to run many simulations of a large farm. As such, many questions remain around designing wake steering strategies at the plant level. Additionally, the physics of wake steering at this scale are not fully understood. In papers such as Medici and Alfredsson (2006), Howland et al. (2016), and Vollmer et al. (2016) it is observed that a pair of counter-rotating vortices are generated in wake steering and that these vortices deform the shape of the wake over time. However, it was not immediately apparent if accounting for these wakes in engineering models was necessary. Some recent research suggests it could be (Fleming et al., 2018b, a).

First, Ciri et al. (2018) note that the scale of the vortices have implications for wake steering's effectiveness, which can be determined by the scale of the rotor to atmospheric scales. They state that larger turbines induce larger wake deflections due to the larger rotor's effects on the length and time scales of the vortex structures. Specifically, larger coherent structures dissipate slower, thus having a longer lasting effect on the wake.

Fleming et al. (2018a) used LES simulations to demonstrate that the vortices lead to discrepancies in existing models of wake steering behavior for arrays of turbines larger than two. One effect, named "secondary steering", shows that the wake of an aligned turbine will be deflected if it overlaps with a steered wake. This was not accounted for in engineering models available at the time the paper was written, and most likely could be explained by vortex interactions.

Very recent papers look at wake steering for larger arrays of turbines. In Bastankhah and Porté-Agel (2019) the authors use scaled wind turbines in a wind tunnel to investigate the optimal distribution of yaw angles for an array of 3–5 evenly spaced turbines. The authors consider various strategies including only yawing the first turbine, or yawing all but the final turbine an equal amount. However, when all but the downstream turbine are free to independently select yaw angles, a more or less linearly decreasing yaw angle assignment from front to rear turbine is found to be optimal.





To determine if wake steering design and analysis could be improved by directly modeling the vortices in a control-oriented model, Martínez-Tossas et al. (2019) propose a theoretical model of vortex-based wake steering. Referred to here as the curled wake model, this model has now been incorporated into the FLORIS control-oriented modeling code as an option for wake modeling alongside the other models referenced earlier.

5     In this paper, we utilize a modified version of the curled wake model in FLORIS (and the FLORIS implementation of Portè-Agel's group, referred to here as the Gaussian wake model). Results for the NREL 5-MW (Jonkman et al., 2009) show that improvements to control design are achieved when using the curled wake model. In the first section, The Gaussian and curled wake models are discussed, including the modifications made to the curled wake model in FLORIS. LES simulations of two and three turbines are then used to indicate the ability of the models to capture wakes from smaller arrays of turbines. We show

10     that the curled wake model is able to predict secondary steering effects. Finally, a three-turbine yaw optimization problem is undertaken to show that more gain in power is possible when targeting these effects. These results are confirmed in LES. The authors note that the curled wake model comparison with the Gaussian wake model is intended to show the value of modeling these additional effects, and not as a detraction from the Gaussian model, which represented a critical step forward in models of wake steering.

15     We believe that this most recent model addition within FLORIS captures important wake effects that need to be considered in wind farm and wake control. The implication of secondary steering on downstream turbines is nontrivial. The results of the new model shown here produce larger energy gains than previously estimated with other wake models. By including these effects in wind plant controller design, we believe this will enable even larger energy gains for larger arrays of turbines.





## 2   FLORIS: A Controls-Oriented Modeling Platform

FLORIS is a controls-oriented wake simulation and wind farm controls analysis tool used to study and optimize wind farm control. It contains several wake models that are computed in steady state and can be used to inform turbine operation as well as wind farm design. FLORIS has been jointly developed by the National Renewable Energy Laboratory (NREL) and Delft Uni-

versity of Technology in the past, and now is available for download and collaborative development (https://github.com/NREL/FLORIS). Below is a brief description of the models in FLORIS, followed by an explanation of the newest model, the curled wake model. For more details on the models in FLORIS, see Annoni et al. (2016). While these are the models included with FLORIS, any wake model can be substituted into the FLORIS framework.

### 2.1   Wake models

FLORIS includes three models: the Jensen model, the FLORIS (multizone) model, and the Gaussian model. The Jensen model (Jensen, 1984) is a well-known, simplified model for calculating wake deficits behind a turbine. The multizone model built upon the Jensen model, as detailed in Gebraad and van Wingerden (2014). This model splits the wake into three different zones: the near-wake zone, the far-wake zone, and the mixing-wake zone. These models, paired with a wake deflection model, such as the Jiménez model (Jiménez et al., 2010), were used in the early development of FLORIS, but all of the recent research

has utilized the Gaussian model. The theoretical basis for the Gaussian model was developed across several papers (Abkar and Porté-Agel, 2015; Bastankhah and Porté-Agel, 2014, 2016; Niayifar and Porté-Agel, 2015; Dilip and Porté-Agel, 2017). The Gaussian model is used in the comparisons in this paper to the new FLORIS model being introduced. The velocity deficit in the wake is described using a Gaussian profile. This model also takes into account added turbulence from the turbines, as well as atmospheric stability.

The Gaussian model uses a simplified derivation of the Navier-Stokes equations to calculate the three-dimensional velocity deficit behind a turbine, defined as:

$$\frac{u(x,y,z,)}{U_\infty} = 1 - Ce^{-(y-\delta)^2/2\sigma_y^2}e^{-(z-z_h)^2/2\sigma_z^2} \qquad (1)$$

$$C = 1 - \sqrt{1 - \frac{(\sigma_{y0}\sigma_{z0})}{\sigma_y\sigma_z}}$$

$$M_0 = C_0\left(2 - C_0\right)$$

$$C_0 = 1 - \sqrt{1 - C_T}.$$

Here $C$ is the velocity deficit at the center of the wake, $\delta$ is the amount of wake deflection, $z_h$ is the turbine's hub height, and $\sigma_y/\sigma_z$ are the width of the wake in the $y/z$ direction, respectively. Each turbine has its own set of these parameters designated with the subscript $i$, which is left out here for conciseness. The initial parameters with subscript $0$ start at the beginning of the far wake, where this location is dependent on the turbulence intensity, $I_0$, and the thrust coefficient of the turbine, $C_t$. For the

definition of the start of the far wake, and additional details, see Bastankhah and Porté-Agel (2016).





The values of the wake widths are calculated as:

$$\frac{\sigma_y}{D} = k_y \frac{(x - x_0)}{D} + \frac{\sigma_{y0}}{D}, \quad \text{where} \quad \frac{\sigma_y}{d} = \frac{1}{2}\sqrt{\frac{u_R}{u_\infty + u_0}}, \tag{2}$$

$$\frac{\sigma_z}{D} = k_z \frac{(x - x_0)}{D} + \frac{\sigma_{z0}}{D}, \quad \text{where} \quad \frac{\sigma_z}{d} = \frac{1}{2}\sqrt{\frac{u_R}{u_\infty + u_0}}, \tag{3}$$

where $k_y$ is a coefficient for the wake expansion in the $y$ direction and $k_z$ is the coefficient for the wake expansion in the $z$
direction. It has been shown that the wake can expand at different rates in the y and z direction, depending upon wake mean-
dering (Abkar and Porté-Agel, 2015). For this work, the two expansion rates have been set to be equal. For the implementation
in FLORIS, the wakes of the Gaussian model are combined using the sum of squares method (Katić et al., 1986). The Gaussian
model also includes relationships that take into account wind shear, wind veer, and turbulence intensity from turbines; however,
the reader is referred to Annoni et al. (2016) for further details on these equations.

## 2.2   Curled wake with vortex decay model

The curled wake model described here is an advancement of the model first proposed in Martínez-Tossas et al. (2019). The
reader is referred to Martínez-Tossas et al. (2019) for a detailed derivation of the model. This model is based on a phenomenon
in turbine wakes identified by Howland et al. (2016), Vollmer et al. (2016), Bastankhah and Porté-Agel (2016), and Medici
and Alfredsson (2006). This phenomenon is observed as a curling of the wake i.e., deforming the initially circular wake of the
rotor to a kidney shape. In addition to this deformation, the wake is displaced laterally. This is due to counter-rotating vortices
that are shed from the rotor when it is not aligned with the wind.

This mechanism of counter-rotating vortices has been confirmed in other work. Bastankhah and Porté-Agel (2016) performed
analyses leveraging both potential flow theory and experiments with a scaled wind turbine that used particle image velocimetry.
Their work identified a pair of counter-rotating vortices. Shapiro et al. (2018) observed the counter-rotating vortices as well,
and determined that the distribution of vorticity from the rotor had an elliptic shape instead of the pair of vortices previously
proposed. This elliptic distribution is important to include, as detailed in Martínez-Tossas et al. (2019) and Shapiro et al. (2018).
Vollmer et al. (2016) also observed the curled wake effect under varying atmospheric conditions in LES.

The model is a simplified and linearized version of the Reynolds-averaged Navier-Stokes streamwise momentum equation
for incompressible flow. The simplifications lead to:

$$U\frac{\partial u'}{\partial x} + V\frac{\partial u'}{\partial y} + W\frac{\partial (u')}{\partial z} = \nu_{\text{eff}} \left( \frac{\partial^2 u'}{\partial x^2} + \frac{\partial^2 u'}{\partial y^2} + \frac{\partial^2 u'}{\partial z^2} \right) \tag{4}$$

which describes the evolution of the wake deficit, $u'$, as it moves downstream. $U$, $V$, and $W$ are the streamwise, spanwise, ver-
tical velocity components from the base flow while $u'$, $v'$, and $w'$ are perturbation velocities around the base flow. Specifically,
the perturbation velocities represent the wake deficits that are convected by the base flow. $\nu_{\text{eff}}$ is the effective viscosity of the
flow.

In order to add the curled wake effect, an elliptic distribution of vortices is added to the base flow (Martínez-Tossas et al.,
2019; Shapiro et al., 2018). The superposition of these vortices with the base flow generates the curled wake shape. The vortices





are individually described as a Lamb-Oseen vortex with a tangential velocity of:

$$u_t = \frac{\Gamma}{2\pi r}\left(1 - \exp\left(\frac{-r^2}{4\nu_{\mathrm{eff}}t + r_0^2}\right)\right), \tag{5}$$

where $u_t$ is the tangential component of the velocity, $\Gamma$ is the vortex (or circulation) strength, $r$ is the radial distance from the core of the vortex, $r_0^2$ is the initial vortex core radius, and $t$ is time. Time here is the time it takes for the flow to travel

downstream, which can be approximated as $t = x/U_\infty$. The curled wake model as presented in Martínez-Tossas et al. (2019) has vortices that do not decay as they move downstream. Here, vortex decay is included in the curled wake model as a scaling factor dependent on the distance downstream from the vortex's formation, based on the work in Bay et al. (2019).

The ratio of the initial vortex velocity to the velocity of a vortex at some later time $t$ can be written as:

$$\frac{u_t(t)}{u_t(0)} = \frac{1 - \exp\left(-\frac{r^2}{4\nu_{\mathrm{eff}}t + r_0^2}\right)}{1 - \exp\left(-\frac{r^2}{r_0^2}\right)}. \tag{6}$$

In its current form, Eq. 6 is dependent on the radial position $r$ within the vortex, and would need to be computed for each turbine at each spatial step downstream. In order to reduce the computational burden of the model, an approximation of this relationship can be made and used to decay the vortices. Expanding the exponential values using their Taylor series expansions and keeping the first two terms of the series gives:

$$\frac{u_t(t)}{u_t(0)} = \frac{1 - 1 + \frac{r^2}{4\nu_{\mathrm{eff}}t + r_0^2}}{1 - 1 + \frac{r^2}{r_0^2}} \tag{7}$$

which, simplifying and substituting $t = x/U_\infty$ arrives at:

$$\frac{u_t(t)}{u_t(0)} = \frac{r_0^2}{4\nu_{\mathrm{eff}}\frac{x}{U_\infty} + r_0^2}. \tag{8}$$

This decay factor is then applied to the vortex velocities, $V$ and $W$, starting at each turbine and marching through the downstream wake.

In addition to the vortex decay factor, modifications were made to the curled wake model presented in Martínez-Tossas et al.

(2019) to account for turbulence added to the flow from turbines. Specifically, the effective viscosity, $\nu_{\mathrm{eff}}$, term in Eq. 4, which is represented as a mixing length model, was modified by the turbine turbulence model. This essentially changes the length scale of the turbulence generated by the turbine. A turbine turbulence model that determines the amount of turbulence added by a turbine was used from Crespo et al. (1996). The turbulence model implemented is defined as:

$$I_+ = 0.73a^{0.8}I_0^{0.1}\frac{x}{d}^{-0.275}. \tag{9}$$

The empirical values used in this model vary from those originally proposed by Crespo et al. (1996). The previously mentioned values have been determined from previous FLORIS studies, as this same turbine turbulence model is used in the FLORIS Gaussian model. This turbulence factor, $I_+$, was applied to the effective viscosity term in Eq. 4, along with a tunable dissipation scaling parameter $\alpha$, as shown in Eq. 10.

$$U\frac{\partial u'}{\partial x} + V\frac{\partial u'}{\partial y} + W\frac{\partial(u')}{\partial z} = \alpha I_+ \nu_{\mathrm{eff}}\left(\frac{\partial^2 u'}{\partial x^2} + \frac{\partial^2 u'}{\partial y^2} + \frac{\partial^2 u'}{\partial z^2}\right) \tag{10}$$





## 3  Validation

For validation of the curled wake model, it is compared against the Gaussian model and SOWFA simulations for both two and three turbine arrays. The turbine used in the simulations is the NREL 5-MW turbine (Jonkman et al., 2009). Both the Gaussian and curled wake models were tuned using SOWFA simulations of a two turbine array. Two sets of tuning parameters were selected, one for a high turbulence intensity (TI) case, and one for a low TI case. The details of the tuning parameters and their values are given in Table 1. It is important to note that there will be differences between the engineering models and the SOWFA data. While near-zero percent accuracy could be achieved, we did not want to overfit the models to these handful of SOWFA simulations, as there can be significant variations between SOWFA cases depending on definition of the inflow, spanwise location of the turbines, etc.

The SOWFA simulations had a mean wind speed of 8.0 m/s, but due to the variations in inflow, the wind speed for the Gaussian and curled wake models was tuned such that the lead turbine produced approximately the same power as the lead SOWFA turbine. These variations in flow are due to the turbulent fluctuations that are seen by the rotor in the SOWFA simulations but are not present in FLORIS. Thus, the tuned wind speed was found to be 8.38 m/s. TI was set at 0.056 for the low TI case and 0.1 for the high TI case, again, based on the SOWFA simulations. The other atmospheric inputs were held constant.

The FLORIS model parameters were then adjusted for each of the wake models until there was reasonable agreement in the relative power differences between the models and the respective SOWFA simulations. Results of this tuning process are described in the next sections.

**Table 1.** Atmospheric and model parameters used in wake model comparisons.

| Atmospheric inputs | Low TI | High TI | Description |
|---|---|---|---|
| Wind speed (m/s) | 8.38 | 8.38 | The freestream velocity of the flow field. |
| Wind direction (deg) | 270.0 | 270.0 | The direction of the freestream velocity of the flow field. |
| TI | 0.056 | 0.1 | A measure of the turbulence intensity present within the flow field. |
| Shear | 0.12 | 0.12 | The coefficient of shear. |
| FLORIS - Curl with decay model parameters | | | |
| Dissipation ($\alpha$) | 0.04 | 0.06 | A tunable parameter that is applied to the flow viscosity in the curled wake model. |
| FLORIS - Gauss model parameters | | | |
| ka | 0.39 | 0.279 | A wake expansion parameter originally derived from empirical data (Niayifar and Porté-Agel (2015)). |
| kb | 0.002 | 0.002 | A wake expansion parameter originally derived from empirical data (Niayifar and Porté-Agel (2015)). |





**Table 2.** Results of the tuning process compared to the respective SOWFA simulations.

| Model | T1 Yaw | T2 Yaw | Difference in Power from SOWFA Low TI | Difference in Power from SOWFA High TI |
|-------|--------|--------|--------------------------------------|---------------------------------------|
| Gauss | 0.0° | 0.0° | -2.26% | -0.11% |
| Gauss | 20.0° | 0.0° | -3.70% | -1.15% |
| Gauss | 0.0° | 20.0° | -2.14% | -0.32% |
| Curl | 0.0° | 0.0° | 3.61% | 4.19% |
| Curl | 20.0° | 0.0° | 2.38% | 2.91% |
| Curl | 0.0° | 20.0° | 3.24% | 3.66% |

## 3.1 Two turbine wake comparison

For the two turbine case, the turbines were spaced 6 rotor diameters (D) apart, as shown in Fig. 1. This is a closer spacing than in some installed wind farms as we believe wake steering can enable tighter spacing of turbines and is of interest for future research. Three yaw cases for each of the TI conditions were used for tuning the models to SOWFA data, giving six simulations of each model. The results are summarized in Table 2. The total power of each case was tuned to +/- 5% of the SOWFA data, which was determined to be sufficiently accurate for control-oriented applications.

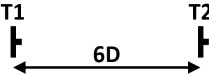

**Figure 1.** Turbine layout and naming used in the two-turbine FLORIS and SOWFA simulations. The turbines are spaced 6D apart.

Overall, the Gaussian model compares well to the SOWFA results when examining turbine powers. This is further bolstered by looking at cross sections of the flow immediately upstream of the second turbine. Fig. 2 displays flow cross sections 5D downstream of the first turbine (or 1D upstream of the second turbine) for the aligned yaw case and yawing the first turbine. Both the Gaussian and curled wake models emulate the wake shape shown in the SOWFA simulations quite well, with some differences in the magnitudes of the velocity deficit (shown by the difference in color). For the curled wake model, this difference in magnitude is a result of balancing the tuning of the model between the baseline and yawed cases. It is part of future work to increase the agreement between SOWFA and the curled wake model with respect to the velocity deficit magnitudes by improving the turbine turbulence model modification and its tuning.

However, as stated in Fleming et al. (2018a), disparities exist between the Gaussian model and SOWFA in the flow downstream of the second turbine. Fleming et al. (2018a) showed that the wake behind the second turbine continues to deflect and deform when the first turbine is positively yawed away from the incoming wind direction, termed secondary steering. Fig. 3 shows a similar depiction, displaying cross sections of the flow 6D behind the second turbine. Both the Gaussian and curled wake model do well in the baseline case with neither turbine yawed, but when the first turbine is yawed, the Gaussian model




fails to predict the continued deflection behind the second turbine. Because of its inclusion of counter-rotating vortices, the curled wake model is able to capture this deflection and the overall shape of the wake as shown in SOWFA.

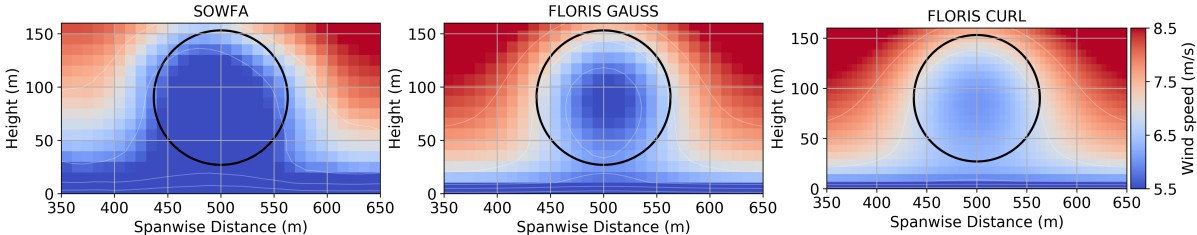

(a) A cross section of the flow 5D downstream of the first turbine with yaw settings of T1= 0.0° and T2= 0.0°.

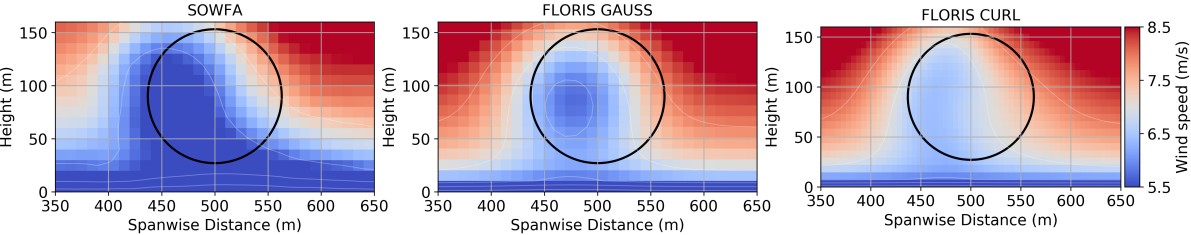

(b) A cross section of the flow 5D downstream of the first turbine with yaw settings of T1= 20.0° and T2= 0.0°.

**Figure 2.** Cross sections of the flow behind turbines for nonyawed and yawed cases.

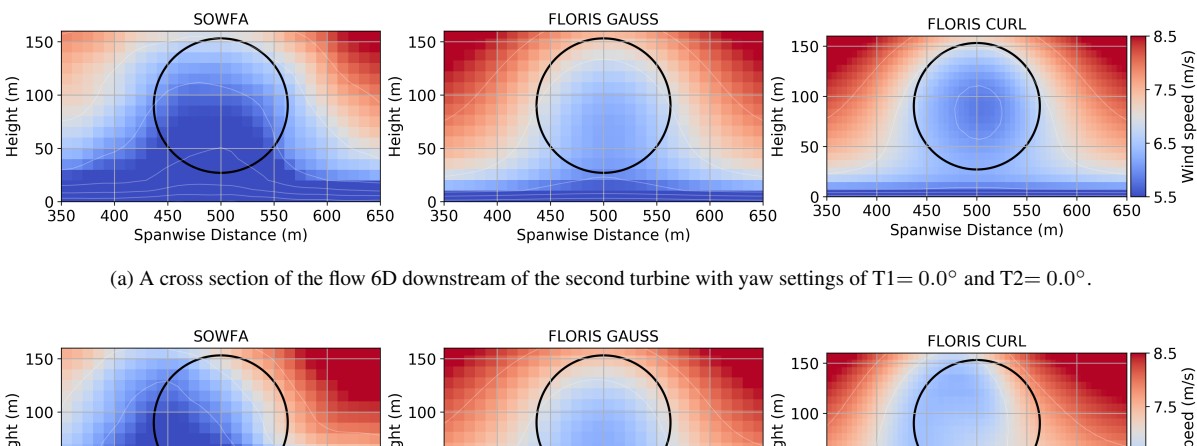

(a) A cross section of the flow 6D downstream of the second turbine with yaw settings of T1= 0.0° and T2= 0.0°.

(b) A cross section of the flow 6D downstream of the second turbine with yaw settings of T1= 20.0° and T2= 0.0°.

**Figure 3.** Cross sections of the flow behind turbines for nonyawed and yawed cases.




To further investigate this difference in wake predictions, the power of a hypothetical third turbine is calculated. This method is inspired by a similar process described in Vollmer et al. (2016). This hypothetical turbine is swept across the flow downstream of the second turbine, as shown in Fig. 4. In the curl plot of Fig. 4, the wake starts in a perpendicular plane to the incoming wind flow. This is an artifact of the marching solution method used to solve the curl model and has little effect on the downstream

5   flow. The power of this hypothetical turbine is calculated across the spanwise distance of the flow at several locations and is compared to the same power calculation performed for the baseline, unyawed case. The results are shown in Fig. 5. The curled wake model follows the power available in the flow from the respective SOWFA simulation much more closely than the Gaussian model. This is because of the counter-rotating vortices and their persistence past the unyawed second turbine. These vortices work to continually deflect the wake of the yawed turbine over distances greater than the normal spacing

10   between turbines. As such, the impact of these vortices becomes even more important when considering anything more than a two-turbine array. This impact on power in a three-turbine array is investigated in the following section.

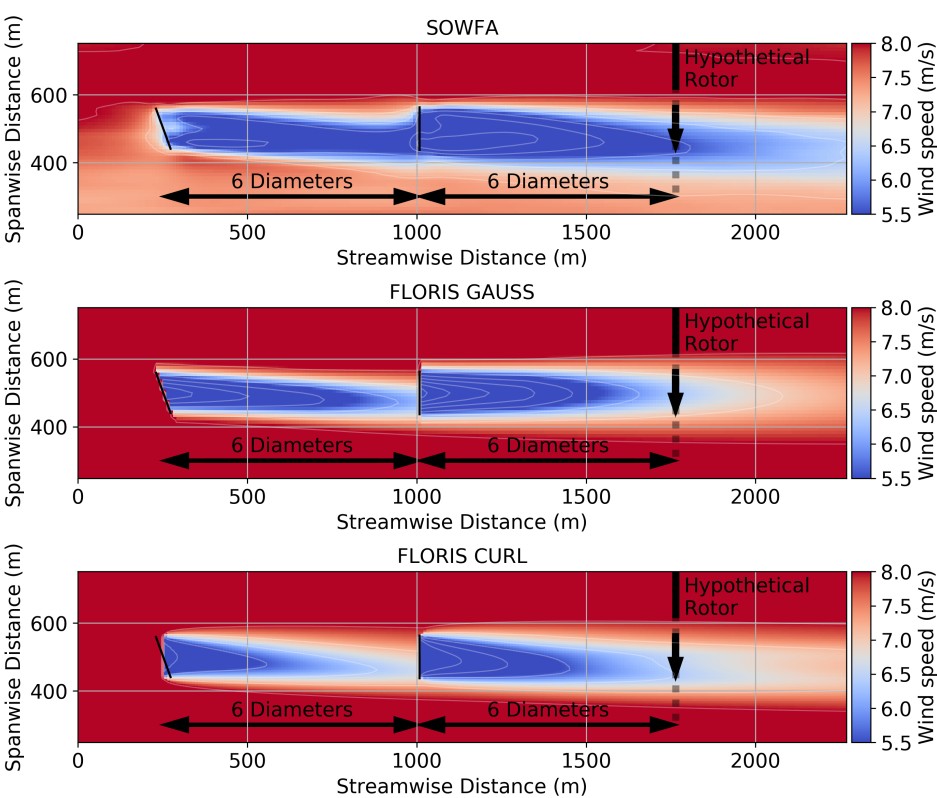

**Figure 4.** Horizontal cross sections of the flow taken at the turbine's hub height of 90 meters. T1 is yawed at 20.0°.




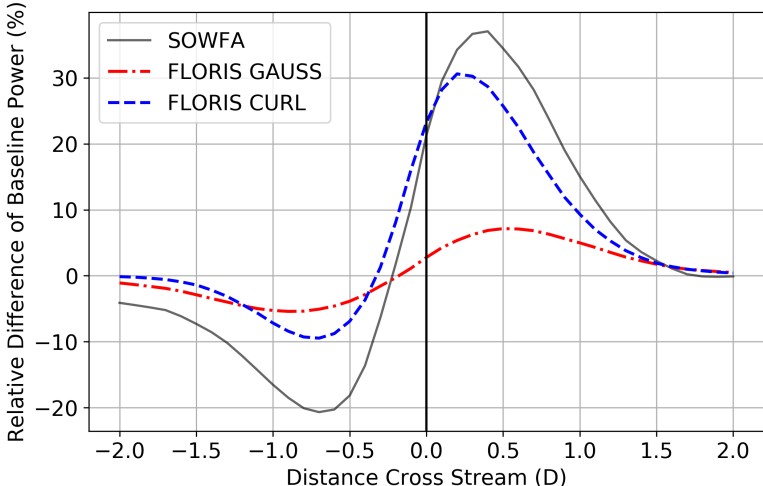

**Figure 5.** Relative change in power of a hypothetical turbine 6D downstream from the second turbine calculated with the FLORIS Gaussian and curled waked models when the front turbine is yawed 20 degrees.

### 3.2 Three-turbine wake comparison

Simulations were performed for a three-turbine case, adding another turbine spaced 6D apart as shown in Fig. 6. The same tuning parameters from the two-turbine case, listed in Table 1, were used for the three-turbine simulations. Overall, the models matched the SOWFA simulations, with a few key exceptions. The Gaussian model suffers in the cases where a yawed turbine is upstream of multiple turbines. On the other hand, the curled wake model has difficulty predicting the baseline, low TI case. This suggests that further tuning and/or improvement is needed in the local turbine turbulence model.

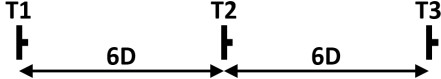

**Figure 6.** Turbine layout and naming used in the three-turbine FLORIS and SOWFA simulations. The turbines are spaced 6D apart.

Examining the flows reiterates the same narrative. For the three-turbine case with the lead turbine yawed 20°, Fig. 7 shows that the curled wake model captures the deflected flow 6D downstream of the third turbine. The Gaussian model does not capture this deflection. Of note is the sustained deflection seen in the SOWFA simulation in Fig. 8. These results bolster the argument that counter-rotating vortices should be included in multiturbine analysis. More importantly, these results illustrate that secondary steering has a significant impact on the performance of wake steering and it is necessary to capture this phenomenon in controls-oriented models.





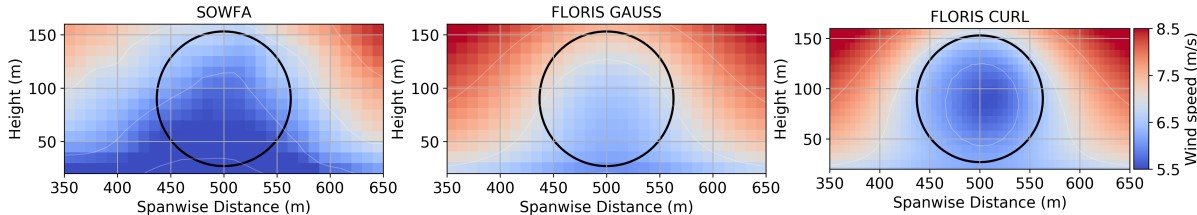

(a) A cross section of the flow 6D downstream of the third turbine with yaw settings of $T1 = 0.0°$, $T2 = 0.0°$, and $T3 = 0.0°$.

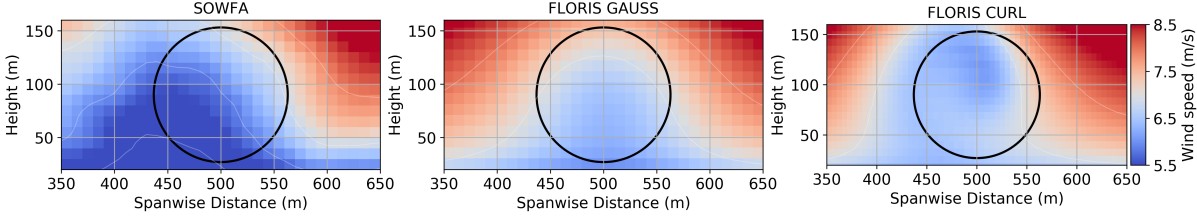

(b) A cross section of the flow 6D downstream of the third turbine with yaw settings of $T1 = 0.0°$, $T2 = 0.0°$, and $T3 = 0.0°$.

**Figure 7.** Cross sections of the flow 6D behind the third turbine for nonyawed and yawed cases.

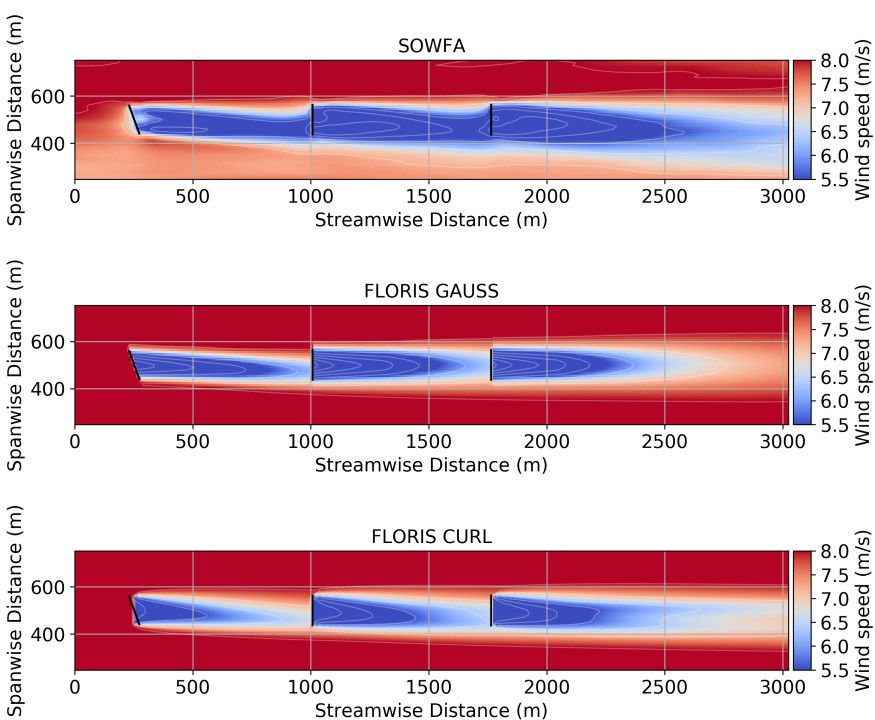

**Figure 8.** Horizontal cross sections of the flow taken at the turbine's hub height of 90 meters. T1 is yawed at $20.0°$.



## 4   Optimization

To further explore the effects of secondary steering on yaw values and wake steering performance, yaw optimizations were run for the low and high TI, three-turbine cases. For both cases, the same environmental and tuning values were used as described in Table 1. The sum of the turbine powers was maximized with the yaw settings of the turbines as the decision variables.

Bounds were set on the yaw angles so they would fall between $0.0°$ and $30.0°$. The SciPy optimization package (Jones et al., 2001–) was leveraged for the optimization, using the SLSQP (Kraft, 1988) minimization method. The objective function of the optimization was defined as:

$$\operatorname*{minimize}_{\gamma} \quad -\sum_{i=1}^{N_t=3} P_i\left(\gamma_i, w_{dir}, w_{spd}\right) \tag{11a}$$

$$\text{subject to} \quad 0.0° < \gamma_i < 30.0° \tag{11b}$$

where $N_t$ is the number of turbines, $P_i$ is the power of a turbine, $\gamma_i$ is the yaw angle of a turbine, $w_{dir}$ is the wind direction, and $w_{spd}$ is the wind speed. The results of the four optimizations (low and high TI performed using the Gaussian and Curl models) are shown in Table 3. For each optimization, a SOWFA simulation was run using the resultant optimized yaw angles. These results are shown below the predicted outputs from the respective wake model. Greater gains were seen for both models in the low TI case, which follows recent results showing that wake steering is more effective in lower turbulence conditions (Vollmer

et al., 2016). The experimental data in Fleming et al. (2019) when binned into lower TI, stable conditions, and higher TI, less stable conditions, show that the gains in the stable case are clearly greater. The Gaussian model predicts these results as well due to the inclusion of TI's effects on recovery and wake deflection.

In both cases, applying the yaw angles determined by the curled wake model optimizations to turbines in SOWFA simulations produced larger power increases than those of the Gaussian wake model. Of particular interest is comparing the magnitude of

the yaw values between the two models. The Gaussian model produces yaw values that increase in magnitude as you progress to downstream turbines; conversely, the curled wake model gives yaw values that decrease in magnitude for downstream turbines. This difference can be explained by the persistence of the counter-rotating vortices and their continual deflection of the wake past downstream turbines. Thus when there are multiple turbines that can yaw, the downstream turbines can add to the superstructure of the wake vortices with a smaller yaw angle, finding the balance between additional power gain at further

downstream turbines and power loss from being misaligned to the incoming wind direction. This model of accumulation of wake steering we believe is a critical contribution and will increase in importance with larger arrays on to full farms.

This same trend was recently published by Bastankhah and Porté-Agel (2019). Bastankhah and Porté-Agel (2019) studied the performance of a five-turbine array in a wind tunnel under various yaw distributions. They found that the yaw strategies that provided the greatest increase in power from the nonyawed baseline (as high as 17%) had a relatively large yaw angle on the

lead turbine, followed by decreasing yaw angles among downstream turbines, finally reaching zero yaw for the rear turbine. Of particular note with regard to the curled wake optimization results in Table 3, the yaw values are close to the mean values from the three-turbine array case in Bastankhah and Porté-Agel (2019) that provide the highest power increase (T1$\approx 27°$, T2$\approx 16°$, T3$= 0°$).





**Table 3.** Yaw optimization performance of the FLORIS Gauss and FLORIS Curl models.

| Model | T1 Yaw | T2 Yaw | T3 Yaw | Optimized Power | Increase in Power Over No Yaw* |
|---|---|---|---|---|---|
| Low Turbulence Intensity | | | | | |
| Gaussian Yaw Optimization | | | | | |
| FLORIS Gauss | 25.2 | 26.1 | 0.0 | 3.65 MW | 8.67% |
| SOWFA | 25.2 | 26.1 | 0.0 | 3.96 MW | 17.45% |
| Curl Yaw Optimization | | | | | |
| FLORIS Curl | 28.7 | 19.9 | 0.0 | 4.02 MW | 24.02% |
| SOWFA | 28.7 | 19.9 | 0.0 | 4.03 MW | 19.39% |
| High Turbulence Intensity | | | | | |
| Gaussian Yaw Optimization | | | | | |
| FLORIS Gauss | 18.9 | 21.7 | 0.0 | 3.80 MW | 2.26% |
| SOWFA | 18.9 | 21.7 | 0.0 | 3.88 MW | 6.52% |
| Curl Yaw Optimization | | | | | |
| FLORIS Curl | 25.1 | 17.1 | 0.0 | 4.08 MW | 9.74% |
| SOWFA | 25.1 | 17.1 | 0.0 | 3.93 MW | 7.88% |

*The percent difference for each model was calculated with respect to that model's baseline power with each turbine at 0.0° yaw.

To illustrate this finding further, sweeps of yaw angles were performed. The lead turbine was set to 25.0° yaw, and the second turbine was yawed from −30.0° to 30.0°. Fig. 9a shows how the power of the third turbine changes due to the shifting yaw of the second turbine and Fig. 9b shows the change in total farm power as turbine two's yaw changes. For the Gaussian model, it appears as if turbine two can yaw positively or negatively and see a gain in farm power. However, according to LES simulations, this is not the case, as shown by the results of the SOWFA simulation.

Specifically, to increase farm power, the second turbine must yaw in the same direction as the upstream turbine (positively in this case). This again supports the notion that these counter-rotating vortices persist past downstream turbines, and as such the downstream turbines can benefit the farm by working with the already deflected wake and not against it. This is immensely important for yaw optimizations of large wind parks and directly counters some previous yaw optimization results. It can be seen that the curl model captures this overall trend, and thus the curled wake yaw optimization produced better results.

To fully explore this result, further yaw sweeps were performed over the entire yaw envelope of the first and second turbines in the two engineering models. The resulting power maps are shown in Fig. 10. The x-axis of the plots is the yaw setting of the first turbine while the y-axis is the yaw setting of the second turbine. The power indicated by the color varying from blue to



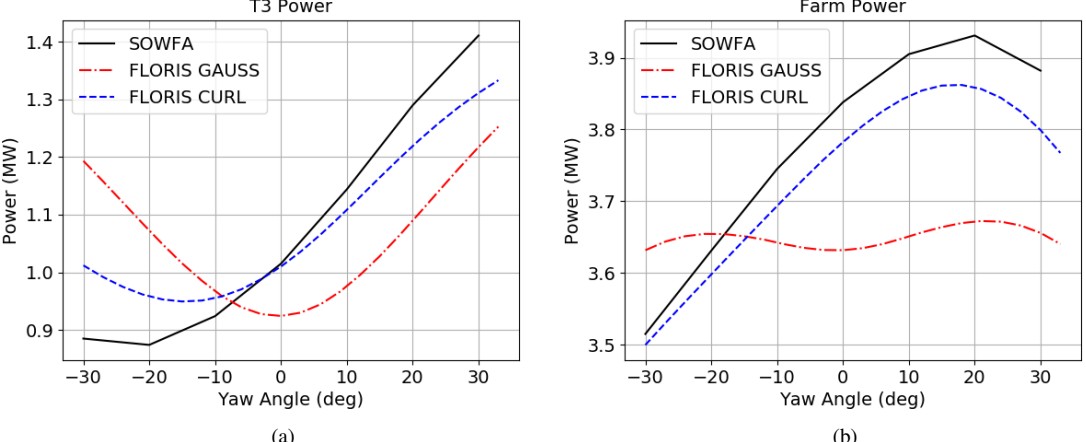

(a)                                                            (b)

**Figure 9.** A plot of power with T1$= 25°$ as T2 is swept from $-30°$ to $30°$.

yellow represents the total farm power. The diagonal red line shows the power where the first and second turbine yaw settings are equal to one another. As such, above the red diagonal line indicates that the first turbine has a greater yaw angle with respect to the second turbine, and below the red diagonal line is the opposite case. The optimal yaw values for the first and second turbines are shown as red pluses.

5    Again, the Gaussian wake model finds that optimal yaw values have the second turbine with a greater yaw angle, whereas the curled wake model finds the lead turbine to have the greater yaw angle. Furthermore, the curled wake model's yaw values predict a larger power increase. Moving forward, the inclusion of these counter-rotating vortices in the controls-oriented models used for controller development and wake steering optimizations/studies is imperative.

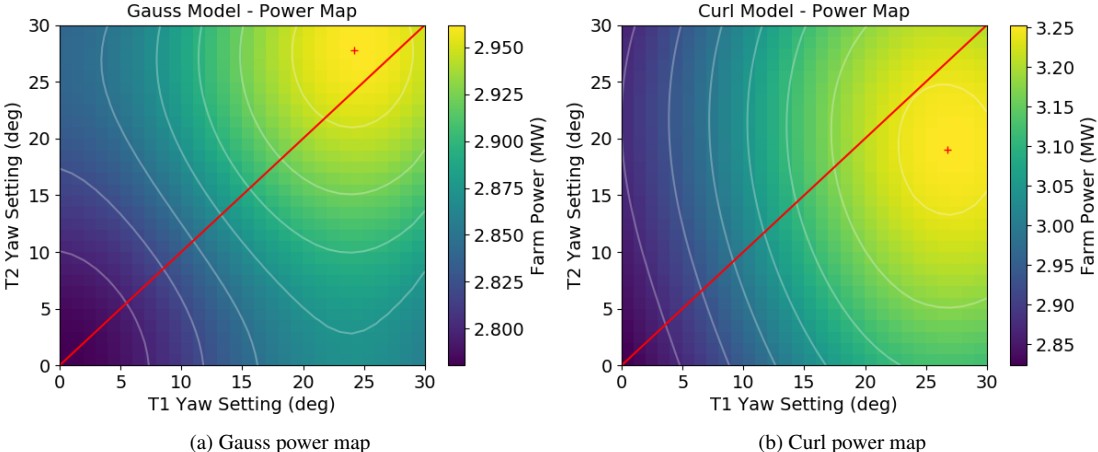

(a) Gauss power map                                   (b) Curl power map

**Figure 10.** Power maps of the Gaussian and curl models across the range of $0°$ to $30°$ for T1 and T2. The red diagonal line represents both T1 and T2 having the same yaw angle. The red plusses indicate the yaw values that produce the greatest power for each model.




## 5    Conclusions

In this paper, a new curled wake model was improved and compared to the existing Gaussian model within the FLORIS software repository and respective high-fidelity SOWFA data sets. This new curled wake model features counter-rotating vortices within the flow structures of yawed turbines. Recent research has shown that these vortices are important to include
in turbine wake control design and investigation. This new curled wake model was shown to capture the secondary steering effects present in both high and low turbulence intensity SOWFA simulations. The addition of vortex decay as well as a local turbulence model to account for turbulence added by operational turbines allowed the new curled wake model to accurately predict wake steering performance for two- and three-turbine arrays.

One important result of the work presented here is that with the new curled wake model, we were able to achieve higher
energy gains than previously predicted with other engineering models. Furthermore, we anticipate that these gains can be larger for larger arrays of turbines. Another significant result of the paper are the yaw angles found from the wake steering optimization. Specifically, the fact that the new curled wake model found yaw angles that progressively decreased in magnitude for turbines further downstream. This is opposite of what the Gaussian wake model optimizations produces; however, this progressive decrease in yaw angle coincides with recent wind tunnel results, also finding that decreases in yaw angle lead to an
increase in performance Bastankhah and Porté-Agel (2019). From this result, a conclusion can be drawn that as a yawed turbine generates counter-rotating vortices that persist beyond downstream turbines, the ideal strategy to improve overall performance is for the downstream turbines to work with, and not against the upstream turbine. Because of these actualities, it is imperative that future yaw optimization studies involving more than a few turbines capture the curled wake effect.

Development of the curled wake model is continuing at NREL, including refinement of the local turbulence model, validation
of the curled wake model in predicting deep array effects, and investigation of optimized wind plant control strategies for large wind power plants leveraging the new controls-oriented curled wake model. Furthermore, comparison of the curled wake model to additional SOWFA cases and refinement of the tuning process is in progress, with the aim of increasing agreement between predictions of each. Accounting for the curled wake phenomenon has the potential to unlock even more performance gains through wake steering, increasing renewable energy production and decreasing the overall cost of wind energy. Future work
will also explore the interaction of curl effects with other atmospheric phenomena, such as wind veer and the possibility of energy entrainment from flow over the farm.



*Acknowledgements.* This work was authored by the National Renewable Energy Laboratory, operated by Alliance for Sustainable Energy, LLC, for the U.S. Department of Energy (DOE) under Contract No. DE-AC36-08GO28308. Funding provided by the U.S. Department of Energy Office of Energy Efficiency and Renewable Energy Wind Energy Technologies Office. The views expressed in the article do not necessarily represent the views of the DOE or the U.S. Government. The U.S. Government retains and the publisher, by accepting the
5    article for publication, acknowledges that the U.S. Government retains a nonexclusive, paid-up, irrevocable, worldwide license to publish or reproduce the published form of this work, or allow others to do so, for U.S. Government purposes.

The authors would like to acknowledge the conversations and feedback received from Eric Simley and Mike Lawson at the National Wind Technology Center.



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
