# Peer review of "Unlocking the Full Potential of Wake Steering: Implementation and Assessment of a Controls-Oriented Model"

_Wind Energy Science, 2019_

## Referee Comment (RC1) · Anonymous Referee #1 · 27 May 2019

The authors present their work on "Unlocking the Full Potential of Wake Steering: Implementation and Assessment of a Controls-oriented Model", where they discuss the addition of the curled wake model from Martinez-Tossas to the FLORIS model, and apply it to demonstration cases using two and three turbines. Overall I think the paper is well written, and I enjoyed reading it. Further, I am happy to see that the FLORIS model has been open-sourced and published on Github, and that the authors keep working on improving the model. However, I feel the paper can be improved significantly in certain areas, which will be discussed point-by-point in my comments below.

**Major comments**

1. I think the title of the manuscript could be improved. The main contribution of the manuscript is on the inclusion of a wake curling model into FLORIS, yet this is not at all reflected in the title. I can understand that the authors might want to have a general title such that this is the reference work to be cited for the FLORIS software, but I feel that currently there is somewhat of a discrepancy between the paper content (which focuses mostly on the addition of the wake curling) and the title (which makes no mention of this at all).

2. Some comments on literature

   (a) The first sentence of the paper briefly introduces the concept of flow control in wind farms. A reference to a review paper such as Boersma et al, "A tutorial on control-oriented modeling and control of wind farms", ACC 2017; or Knudsen, Bak and Svenstrup, "Survey of wind farm control -power and fatigue optimization", Wind Energy 2015 would be appropriate.

   (b) The concept of wind-farm control is viewed here from a power maximization perspective, but obviously loading is important as well. Some comments on the loading of yawed wind turbines, or wind turbines in partial wakes (e.g. due to partial upstream wake deflection, see Figure 4) would be relevant.

   (c) I think the work of Munters and Meyers, "Dynamic Strategies for Yaw and Induction Control of Wind Farms Based on Large-Eddy Simulation and Optimization", Energies 2018 11(1), would be relevant. In this work, they use gradient-based optimization using LES directly as a control model, hence inherently accounting for wake curling. They show that the optimizer exploits the curling behavior by curling wakes just around subsequent turbines. This could be added as additional evidence that incorporating curling effects in control models is necessary (e.g. on p. 2, line 22).

Furthermore, for an aligned set of 4 turbines, they observe yaw angles of 37 deg, 17 deg, and 17 deg for the first, second and third row. These values lie close to the ones observed by Bastankhah and Porte-Agel, cited around line 30 of page 13. Furthermore, a follow-up paper by Munters and Meyers, "Optimal dynamic induction and yaw control of wind farms: effects of turbine spacing and layout", J Phys Conf Series 2018, performed for much larger wind farms, shows similar behavior to the one observed by the Curl model, i.e. decreasing yaw angles as you progress to downstream.

I think both these papers, which design control laws using a high-fidelity model directly, nicely align with the observations shown here by the Curl model.

3. I have several comments on the validation section

   (a) The details of the SOWFA reference setup should be elaborated more, this is important from a reproducibility context. For instance: what turbine model is used, are buoyancy and Coriolis effects included, what is the grid resolution, how long has the LES been time-averaged, ...

   (b) Section 3 starts with the sentence, "For validation of the curled wake model, it is compared against the Gaussian model ...". It is not immediately clear from this sentence what you compare exactly. Later it becomes apparent that it is the turbine power you're comparing (and not e.g. wake deflection, wake deficit, ...), maybe it's best to explicitly mention this in the opening sentence, as this only became clear to me when looking at Table 2.

   (c) "While near-zero percent accuracy could be achieved, we did not want to overfit the models..." This is a vague statement. Firstly, you probably mean near-zero percent error, or near-hundred procent accuracy instead of near-zero percent accuracy. Secondly, how did you define a threshold for how well your models should be trained? Later, on page 8, line 6 you indicate

that you tune up to 5% error, maybe you should also mention this 5% on page 7.

(d) Figure 2 and 3 have identical captions, please update them to be more specific.

(e) All contourplots except for Figure 10 use the coolwarm colormap, which is a suitable map for diverging data that has a clear midpoint to show deviations from. However, your data is not centered around a meaningful value (at least not in the current representation, one could argue that the inlet velocity could be a midpoint to show deviations from but this is not the case here), so this diverging representation does not add much. A sequential colormap, which is intelligible in black & white, would be more suitable in my opinion (e.g. the viridis map you use in Figure 10). Furthermore, you cut the colormap with a minimum value at 5.5, whereas the LES data seems to go below that value in the wake center. Given the qualitative nature of comparing colored contourplots, this can be misleading. I would advise to show the full range of data values. If this range results in less informative plots, then I advise to at least make a small comment about the fact that you're cutting the data below 5.5

(f) Figure 2a. Why is there so much difference between the FLORIS GAUSS model and the FLORIS CURL model? Do they not collapse for non-yawed turbines? Or is this an effect of different tuning parameters for both models? Also, the FLORIS CURL seems to be represented on a much finer grid, why is this?

(g) (suggestion) Maybe it's interesting to show streamlines in the crossplane for Figures 2 and 3 that show the curling motion of the flow

(h) Figure 4. In the LES data, there seems to be quite some shear in the spanwise direction (approx. 10%) Does this have any influence on the tuning of the models? This also seems to be reflected in Figure 5, where the LES

power is lower at -2D than at +2D)

(i) I don't really see the added value of Section 3.2, with the three-turbine wake comparison. The turbine powers are discussed qualitatively in the text, but quantitative results are not given. What does this section have to add over the two turbine wake comparison? If the idea is to test whether the settings, as tuned for two turbines, translate to three turbines, then please indicate this clearly in the text.

4. Page 14, Table 3. It seems that the Gauss model consistently underpredicts the achieved power gains, whereas the Curl model overpredicts it. The former can be explained by the facted that curling deflects the wake centerline more than expected. For the latter however, I have no intuitive explanation of the results. Does the curl model overpredict deflection and/or curl? Could the authors comment on this

5. Figure 10 provides an interesting visualization on the increasing/decreasing yaw angle for the Gaussian/Curl wake models. However, I feel that a similar figure with LES results (albeit at lower resolution yaw angle resolution) would improve this figure and its discussion significantly. For instance, using a 5 deg resolution for T1 and T2 yaw, this could be done by using 49 LES simulations, which should be feasible from a computational perspective. Some of these simulations have already been performed for the data in Figure 9 anyway.

**Minor / typos /...:**

1. (p 3, l 7) The → the

2. (p 5, eq 2 and 3) I'm assuming the expression on the right in both equations should be $\sigma_{y0}/D$ and $\sigma_{z0}/D$ respectively instead of $\sigma_y/d$. Furthermore, $u_R$, $u_\infty$,

and $u_0$ are not explicitly defined.

3. (p 6, l 4): $r_0^2 \rightarrow r_0$

4. (p 10, l 4) "This is an artifact of the marching solution used to solve the curl model". Is a similar marching solution method not used for the Gauss model then?

5. (p 13, l 6) The authors use a SLSQP method for the optimization of the decision variables. How is the gradient calculated? I'm assuming this is through finite differences. Please mention this in the manuscript.

6. Figure 9 (a) and(b). Please use "T2 Yaw Angle" in the x-label over "Yaw Angle"
* * *

---

## Referee Comment (RC2) · Anonymous Referee #2 · 21 Jun 2019

The manuscript presents a modified version of the recently-developed curled wake model. Predictions of this model and those obtained by the Gaussian wake model are compared with SOWFA simulations performed for two and three rows of wind turbines. By highlighting the importance of so-called secondary steering in wind farm yaw angle control, the presented results show the advantage of the curled model over the Gaussian one. Wake steering is an important stream of research since it is arguably deemed as the most promising wake mitigation strategy in the wind energy community. Therefore, I greatly appreciate recent efforts of the authors to develop computationally inexpensive models that can predict wake flows of yawed turbine more realistically.

The curled wake model has been already presented in the previous work of the authors (Martínez-Tossas et al. (2019). Therefore, I can summarize the contribution of the present manuscript as follows:

- Improve the theoretical framework of the curled wake model. This was achieved by adding (i) an additional term that accounts for vortex decay in the wake, and (ii) the effect of added turbulence by the turbine using the relationship developed by Crespo et al. (1996).

- Detailed comparison of curled wake model predictions with those of the Gaussian model to clearly show the higher accuracy of the former model and the importance of secondary steering in wind farm control strategies.

I think the authors did a great job addressing the second point mentioned above. However, I am afraid that I found the first part (theoretical model improvement) relatively sketchy and shallow. Eq. 6 is used to decay the vortex strength as the wake moves downstream. However, no explanation is provided on why this relationship was used. The validity of this relationship can be first verified using SOWFA simulations. Likewise, the turbulence added by the turbine is simply added by multiplying the effective viscosity by I+ and a constant coefficient. Indeed, it can be useful to provide physical insights on the validity of this approach. To sum up, I think this work is important and useful for the wind energy community, but in my opinion the theoretical part needs to to be improved first. Some more specific comments can be found below:

- Page 1, Line 23: Please check again Park et al. (2016). I think it should be "four" instead of "six".

- Page 2, line 3: I could not find Howland et al. (2019).

- Page 2, lines 21-22: Please rephrase this part. It seems that the last sentence is missing some parts.

- Page 4, line 16: If I am not mistaken, the last mentioned reference (Dilip and Porte-

Agel 2017) is not about the Gaussian wake model. Please also reorder the references according to the publication date.

- Eq. 1: M0 is defined but it is not used in the main equation. Please check this.

- Eq. 2 and 3: I cannot fully understand these equations too. It seems that there is another typo as simga_y and sigma_z are both defined two times. Please let me emphasize that as the paper discusses theoretical models, it is extremely important to ensure that all of the equations are written correctly. Otherwise, this may undermine the usefulness of this work for other researchers.

- Eqs. 2 and 3: u_R, u_0 and u_inf are not defined.

- Eq. 4: It is not clear how v_eff is calculated.

- Table 1: Are k_a and k_b the same as k_y and k_z? Why do they have so different values?

---

## Author Comment (AC1) · 24 Sep 2019

The authors would like to thank the reviewers for their comments and time spent reviewing our paper. We have been working diligently to address the reviewer's comments and to update our results based on some recent changes/fixes to the FLORIS wake simulation code. During this process, as we have investigated some of the details requested by the reviewers and dug into some of the details behind the turbulence model, we have identified that the curled wake model can be tuned to match several cases very well, but as implemented, these tunings do not cover the full range of environmental conditions without additional retuning. Through these efforts, we have found

areas of the implementation of the turbulence that we plan to improve over the next 6-12 months.

While our initial intention with this paper was to quickly publish the preliminary results and the model details with the open-source code so that users could have a reference (which was accomplished with the record available on Wind Energy Science Discussions), we are now more of the mind of taking additional time to refine the implementation, further validate the model predictions, and more fully address the range of turbulence and wind conditions that can exist for the final version of the paper. As such, we will withdraw this current paper and re-submit an updated version in the near future, allowing us to fully address the reviewer's comments and implement the additional improvements that we have decided to pursue.

Again, thank you for your time and effort. Your comments and feedback help us to publish the best research possible.

―――――――――――